# Active metabolism unmasks functional protein–protein interactions in real time in-cell NMR

Leonard Breindel[1], David S. Burz[1] & Alexander Shekhtman [1]✉

Protein–protein interactions, PPIs, underlie most cellular processes, but many PPIs depend on a particular metabolic state that can only be observed in live, actively metabolizing cells. Real time in-cell NMR spectroscopy, RT-NMR, utilizes a bioreactor to maintain cells in an active metabolic state. Improvement in bioreactor technology maintains ATP levels at >95% for up to 24 hours, enabling protein overexpression and a previously undetected interaction between prokaryotic ubiquitin-like protein, Pup, and mycobacterial proteasomal ATPase, Mpa, to be detected. Singular value decomposition, SVD, of the NMR spectra collected over the course of Mpa overexpression easily identified the PPIs despite the large variation in background signals due to the highly active metabolome.

[1] Department of Chemistry, University at Albany, State University of New York, 1400 Washington Ave, Albany, NY 12222, USA.
✉email: ashekhtman@albany.edu

Characterizing functional protein–protein interactions, PPIs, within a living cell is essential for understanding the origins of biological activity[1–3]. PPIs underlie most cellular processes and have become popular albeit challenging drug targets. Part of the challenge is due to the fact that full activity arising from PPIs may depend on a particular metabolic state that can only be achieved in live, actively metabolizing cells where the interacting surfaces are subject to holistic regulation by the metabolome and its byproducts[4,5]. In-cell NMR spectroscopy partially overcomes this problem by collecting NMR spectra on protein targets labeled with NMR active nuclei, primarily [15]N and [13]C, inside living cells[6–11].

Maintaining active metabolism and a high degree of cell viability remain major impediments[12–14] to most in-cell NMR work. Cell death is common during prolonged experiments and controls are exhaustively employed to insure that signals arise from within intact cells. In-cell NMR spectra collected using quiescent or metabolically inactive cells do not accurately reflect the effect of the interaction network, interactome, on the target molecule under study[12,15,16]. The introduction of bioreactors[12–14,17,18], which continually introduce fresh growth medium and remove waste material, help alleviate this problem by establishing conditions more favorable to cell growth.

The original design of our bioreactor sustained metabolic energy levels in E.coli for up to 24 h with ~40% loss of ATP, ADP, NAD[+], and NAD(H) concentrations, consistent with >90% cell viability and limited cell growth[13]. However, continuous high levels of energy charge are required for cells to sustain full metabolic processes such as transcription, translation, replication and other ATP-dependent processes. The improvements presented in this study sustain high levels of ATP in E. coli sufficient for protein expression and vigorous growth. By maintaining a highly active cellular metabolism ATP-dependent functional PPIs can be identified by combining real time, RT, and structural interactions, STINT[8], in-cell NMR[13].

The mycobacterial proteasome system[19], was used to illustrate the importance of studying functional PPIs in-cell. In mycobacteria, proteins are targeted for the mycobacterial proteasomal ATPase, Mpa, and the 20S proteasome, by post-translational modification with prokaryotic ubiquitin-like protein, Pup. Pupylation affects up to 5% of the M. tuberculosis, Mtb, proteome[20–22]. The Pup-proteasome system is implicated in persistent infections of macrophages by Mtb and is recognized as an important drug target[23–25].

In this work Pup–Mpa interactions in E. coli are re-examined using an improved version of our bioreactor that maintains high cellular metabolism for up to 24 h sufficient to support vigorous cell growth. E. coli were used to isolate the Pup–Mpa system from undesired Mtb proteasome factors present in the native host. The functional protein–protein interactions that were observed can only occur in the presence of an active metabolome.

## Results

The ability to maintain cells in a metabolically active state is critical for the success of any in-cell NMR study[13]. During in-cell NMR experiments in the absence of a bioreactor, high numbers of cells are suspended in NMR buffer for several hours as NMR spectra are collected. During this time waste is being produced, and the pH and redox state of the cell is changing, which can lead to changes in the NMR spectra as the protein reacts to the cellular environment[12]. Changes in the cellular environment ultimately lead to cell death and leakage of target protein from the cell resulting in NMR spectra that are a combination of in-cell and free protein.

**Improved bioreactor**. A schematic of the improved bioreactor is shown in Fig. 1 and Supplementary Fig. 1. Cells were maintained at 310 K with fresh medium incubated at 315 K to allow for temperature loss during the transfer from the reservoir to the NMR tube. An enhanced system to maximize exposure of cells to fresh medium was developed by replacing the original drip orifices[13] with an ultrahigh molecular weight, UHMW, microporous hydrophilic polyethylene diffuser. The diffuser has pores that range from 50–100 μm and provides a uniform dispersal of the medium over a large, ~63 mm[2], surface area.

To maintain a high concentration of cells within the NMR sampling volume E. coli cells were encapsulated into 1% alginate beads by using an atomizer. Atomization created a highly reproducible uniform dispersion of beads[26] (Fig. 2a). Unlike agarose threads, alginate expands as cells grow. The average diameter of alginate beads cast with E. coli was 0.83 ± 0.03 mm; after 24 h in the bioreactor the diameter had increased to 0.90 ± 0.04 mm, corresponding to a roughly 25% increase in volume (Supplementary Fig. 2). Although some cells are ejected from the matrix, the overall cell density within the sampling volume remains constant, maintaining signal strength. The uniform dispersion of medium provided by the diffuser minimizes the upward drag of the flowing medium on the cast beads preventing them from moving about in the sample volume. Cells ejected from the alginate matrix are washed away by the flow of medium and do not contribute to the spectra.

**Extended cell viability**. Previous work[27] showed that E. coli cell viability can be characterized into three categories based on the adenylate energy charge. At high ATP concentrations cells grow vigorously and exhibit high metabolic activity. Following a growth cycle or during carbon starvation the levels of ATP slowly decrease, retaining cell viability and the ability to form colonies. As the ATP concentration drops even further metabolic inactivity or cell death occurs. The clear delineation of cell vitality makes E. coli a good choice for optimizing in-cell NMR spectroscopy to identify functional PPIs under conditions of maximum metabolic levels and the highest levels of cellular ATP.

The viability of E. coli cells in the bioreactor was monitored with and without the flow of fresh medium (Fig. 2b and c). Because the intracellular concentration of ATP changes as cells enter different stages of growth[28], in the course of an in-cell NMR experiment [31]P spectra were collected to monitor the levels of phosphate-containing metabolites, ATP, ADP, NAD[+], and NAD(H)[18,29]. ATP in particular is a key indicator of metabolic activity. Under flow conditions the vitality of the cells remained steady for 24 h. In the absence of flow phosphate-containing metabolite levels dropped by 10% in 8 h, 30% in 12 h and 40% after a 24 h period (Fig. 2c). Continuous high levels of phosphate-containing metabolites are consistent with a high energy charge and vigorously growing metabolically active cells.

**Protein overexpression**. To demonstrate the high metabolic activity of E. coli in the improved bioreactor protein overexpression was followed in real time. Optimal bioreactor protein overexpression required designing a modified growth medium that contains a minimum amount of phosphate[30]. Phosphate chelation of calcium from the alginate breaks down the encapsulating beads and releases cells into the medium[31]. The hybrid growth medium, HGM, is similar to M9[32] but is buffered with HEPES and supplemented with [U- [15]N] ISOGRO powder for uniformly [15]N labeling target protein. For overexpression of unlabeled protein the ISOGRO is replaced by yeast extract and tryptone.

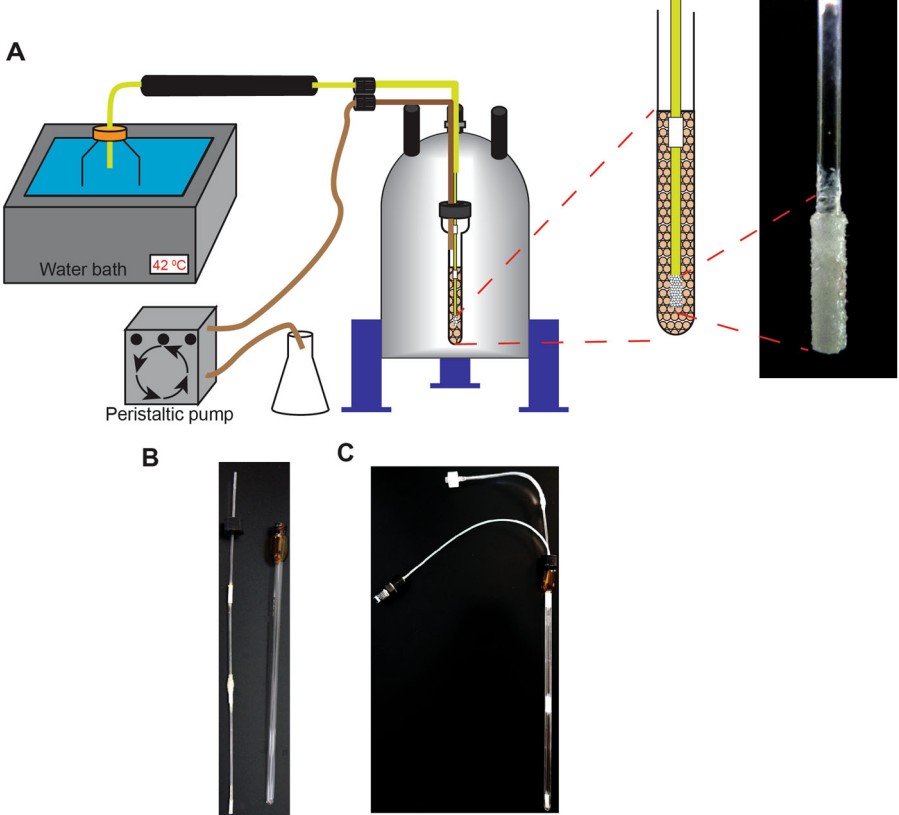

**Fig. 1 Bioreactor setup. a** (Left) A peristaltic pump is used to draw fresh medium through the bioreactor at a rate of 80 μL/min. Samples are maintained at 37 °C in the bioreactor and a water bath is used to keep the medium at 42 °C to compensate for temperature loss during transfer to the NMR tube. (Right) A magnified image of the diffuser tip. **b** Disassembled bioreactor components are shown: A drip irrigation stem and capped NMR tube. **c** The assembled bioreactor.

Bioreactor overexpression of unlabeled mycobacterial proteasomal ATPase, Mpa, was conducted over a 24 h period on the benchtop at 310 K. Samples were removed at 2 h intervals and electrophoresed on a 10% denaturing polyacrylamide gel (Fig. 3a). The concentration of Mpa increased steadily for 12 h, and decreased over the final 12 h due to intracellular proteases that degrade overexpressed protein. This experiment demonstrated the ability of the bioreactor to maintain cells in a high metabolic state with sufficiently high adenylate charge to sustain the ATP-dependent process of protein overexpression.

Bioreactor overexpression of $[U\text{-}^{15}N]$-labeled prokaryotic ubiquitin-like protein, Pup, inside the NMR spectrometer was monitored by using $^{15}N$-edited band-Selective Optimized Flip Angle Short Transient Heteronuclear Multiple Quantum Coherence, SOFAST-HMQC, NMR (Fig. 3b–e). The spectrum of purified Pup was used to identify in-cell cross-peaks. Maximum intensity of Pup spectral peaks was achieved after 12 h. After 24 h spectral peaks corresponding to all 64 amino acids were still evident. The ability to monitor protein overexpression expands the versatility of in-cell NMR spectroscopy to identify functional PPIs in real time.

**Functional protein–protein interactions**. To demonstrate the ability of the bioreactor to investigate functional protein–protein interactions in real time, the interaction between Pup and Mpa was examined by real time structural interactions, RT-STINT, NMR spectroscopy. Previous characterization of the interaction surface between Pup and Mpa[33,34] obtained in vitro and in metabolically inactive cells, served as a comparative standard for the ability of the bioreactor to overexpress protein and identify

interactions occurring in the cellular environment. STINT-NMR monitors changes in the in-cell NMR a spectrum of an isotopically labeled target protein over time as the concentration of an unlabeled interactor protein increases[8]. To perform RT-STINT NMR protein expression must occur within the bioreactor.

$[U\text{-}^{15}N]$ Pup was overexpressed in *E.coli* in the bioreactor on the bench top and the bioreactor was transferred to the magnet. A $^{15}N$-edited SOFAST-HMQC in-cell spectrum of free Pup was collected and the medium was changed to induce overexpression of unlabeled Mpa. Spectra were collected at hourly intervals during which time minimal cell growth was observed. Previous work showed that the concentration of Pup remains constant during overexpression of Mpa[33]. Crosspeaks associated with residues of Pup that interact with Mpa broadened over time as the concentration of intracellular Mpa increased (Fig. 4a). The intensity values were assembled into a matrix, M, for analysis to identify residues involved in the binding interaction[34].

As a test of the reliability of RT-STINT NMR spectroscopy, the apparent affinity of the Pup–Mpa interaction was estimated (Fig. 4b). Benchtop bioreactor temporal overexpression of Mpa was quantitated. Strong spectral overlap of the in-cell NMR Pup peaks prevented exact quantification of the Mpa–Pup binding for most of the residues. Differential broadening of the I18 crosspeak intensity was correlated with Mpa concentrations based on the rate of benchtop overexpression. Since it was impossible to separate cells that have overexpressed only Pup or Mpa from cells that contain both Mpa and Pup and thus contribute to Pup–Mpa binding, the analysis of binding was at best approximate. The resulting curve was fit with an apparent $K_d$ of 28 ± 3 μM, which is in rough agreement with previous estimates of 3–4 μM[35] (Fig. 4b).

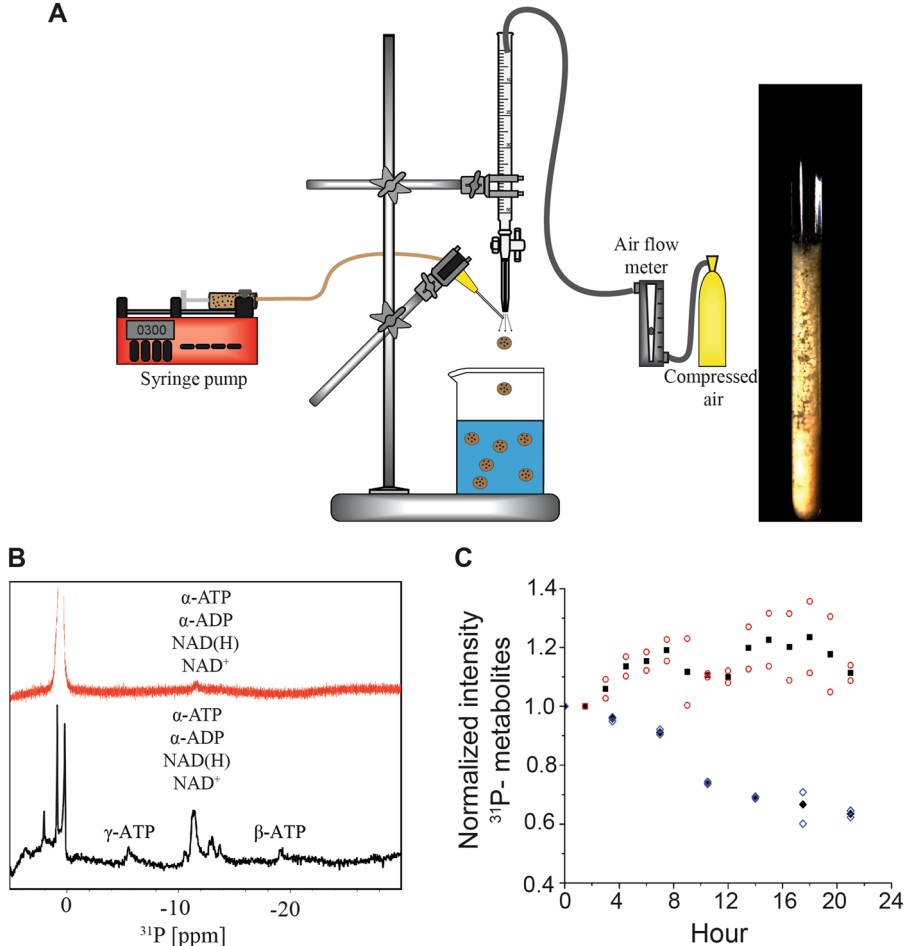

**Fig. 2 Schematic of the cell casting apparatus and time course of the energy charge of cells in the bioreactor. a** An atomizer was used to cast *E. coli* in 1% alginate. 150 mM CaCl$_2$ was used to polymerize alginate into beads upon contact. **b** and **c** Continuous medium exchange maintains high levels of phosphate-containing metabolites. **b** In-cell $^{31}$P spectra of phosphate-containing metabolites in *E. coli* without (red) and with the bioreactor (black). **c** Signal intensity of intracellular $^{31}$P-containing metabolites over time for *E. coli* with (filled squares) and without (filled diamonds) a flow of medium. The experiment was performed in duplicate and all data points for *E. coli* with (open circles) and without (open diamonds) a flow of medium are indicated.

Because of the large variation in signals obtained from vigorously growing cells, distinguishing specific binding interactions from background noise is essential for accurately interpreting in-cell NMR spectroscopic data. Singular value decomposition, SVD, which identifies the principal components of a matrix, M that make the largest contribution to the variance of matrix elements[36,37], was used to analyze the RT-STINT NMR data. SVD identifies residues whose signals correlate with the increase in interactor concentration and not the "noise" associated with changes in metabolic fluxes, and cell division, which dilutes the intracellular concentration of labeled target protein[13]. Binding modes are ranked by their singular values, SV, and plotted in a bar graph Scree plot (Fig. 5a).

Practical selection of true binding events involves linear least squares analysis of the SVs in the Scree plots. A poor linear fit indicates the presence of one or more binding modes; noise (Supplementary Fig. 3) or dilution due to cell growth[13] results in singular values that can be fit to a linear curve with $r^2 > 0.9$. The poor linear fit, $r^2 = 0.61$, of the singular values obtained for the Pup–Mpa interaction showed that the first binding mode constituted a real signal that was distinct from the noise observed in successive binding modes (Fig. 5a). Eliminating the first binding mode for the Pup–Mpa interaction Scree plot and refitting the data produced a good linear correlation, $r^2 = 0.98$ (Fig. 5a).

The abrupt drop in the Scree plot[34] of the Pup–Mpa interaction revealed an interaction surface within the helical section of Pup, residues 20–52 (Fig. 5b, c), comparable to what was previously reported both in vitro and in metabolically inactive cells (Supplementary Fig. 4). The binding included a second previously unobserved set of interacting residues encompassing the N-terminus of Pup (Fig. 5b, c and Supplementary Fig. 4). These interactions implicate an ATP-dependent process previously cited as critical for pupylated substrates to enter the proteasome for degradation (Fig. 5d)[38]. The ability to observe this functional PPI was possible because the improvements in bioreactor technology were able to maintain the cells in a highly metabolic state.

Striebel et al[38]. showed that in the absence of ATP, pupylated substrates bind to Mpa coiled coil domains via the helical region of Pup but do not undergo unfolding or degradation. Mpa cannot unfold Pup and no interaction occurs between the Pup N-terminus and Mpa[38]. The interacting residues in the N-terminus thread Pup through the ATP activated Mpa channel and into the proteasome (Fig. 5d). Previous studies that employed STINT-NMR in metabolically inactive cells to characterize the interaction between Pup in the presence of the *Mycobacterium* proteasome ATPase, Mpa[33,34], were able to visualize the interaction between the C-terminus of Pup and Mpa$_{46-96}$ coiled coil[39] but failed to see the ATP-driven interaction between the N-terminus of Pup with

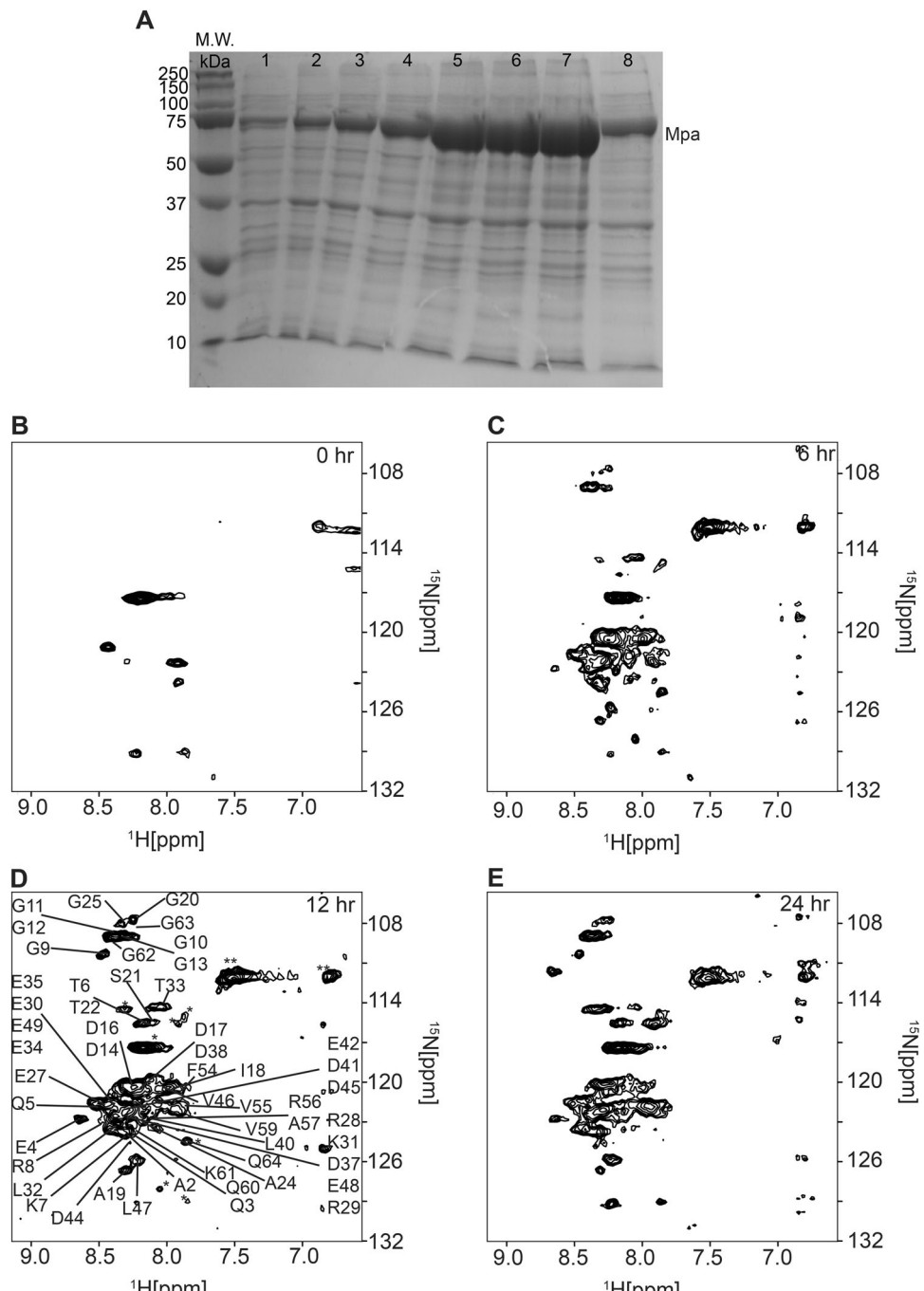

**Fig. 3 Overexpression of Mpa and Pup in the bioreactor. a** SDS-PAGE of Mpa overexpression. Lanes 1-7 are Mpa expression samples collected every 2 h for 12 h, lane 8 is Mpa expression after 24 h. **b**, **c**, **d**, **e** [15]N-edited SOFAST-HMQC spectra of Pup overexpression in the bioreactor during in-cell NMR after 0 (**b**), 6 (**c**), 12 (**d**), and 24 (**e**) hours.

the inner channel of Mpa[38]. The lack of visibility of the N-terminal interaction in the previous studies[33,34] was due to insufficiently high levels of ATP within the cells. Without the bioreactor, ATP levels within the cell rapidly decline making it impossible to detect this interaction during traditional in-cell STINT NMR experiments.

To test the hypothesis that the binding of ATP alone is sufficient to generate the Pup–Mpa interaction, a non-hydrolyzable ATP analog β,γ-methyleneadenosine 5'-triphosphate, AMP-PCP was added to *E. coli* lysates containing overexpressed Pup and purified

Mpa. Adding AMP-PCP did not change the intensities of the N-termini residues, indicating no interaction between Pup and Mpa (Supplementary Fig. 5). This suggests that ATP hydrolysis is required for the Pup N-terminal region to engage with Mpa. Because Mpa driven ATP hydrolysis is very rapid, ~1.2 s$^{-1}$[40], an extremely high non-physiological concentration of ATP, >10 mM, would be required to observe this interaction in vitro over the time course of an NMR experiment. To best visualize such high energy interactions using in-cell NMR, cell health must be maintained for the duration of the NMR experiment.

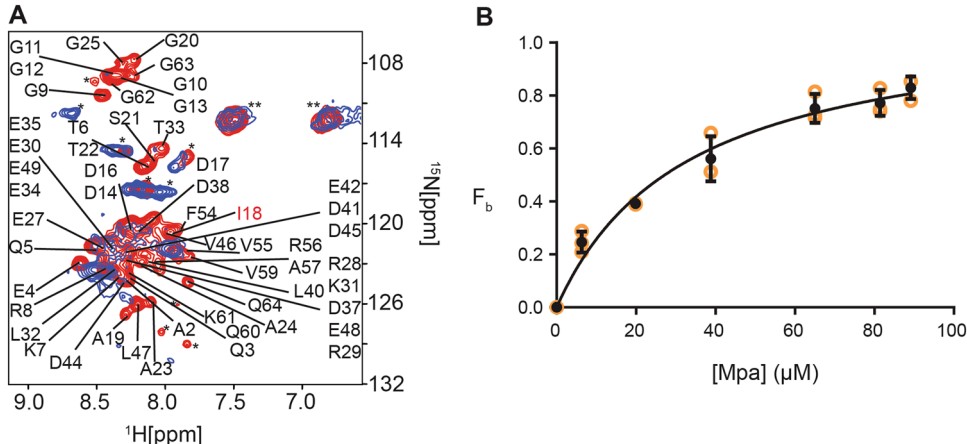

**Fig. 4 RT-STINT NMR of the Pup–Mpa interaction. a** Overlay of $^{15}$N-edited SOFAST-HMQC spectra of Pup before (red) and after (blue) 12 h of Mpa overexpression. **b** Apparent binding isotherm of Pup to Mpa in *E. coli*. The error bars give the standard error of the mean for each data point.

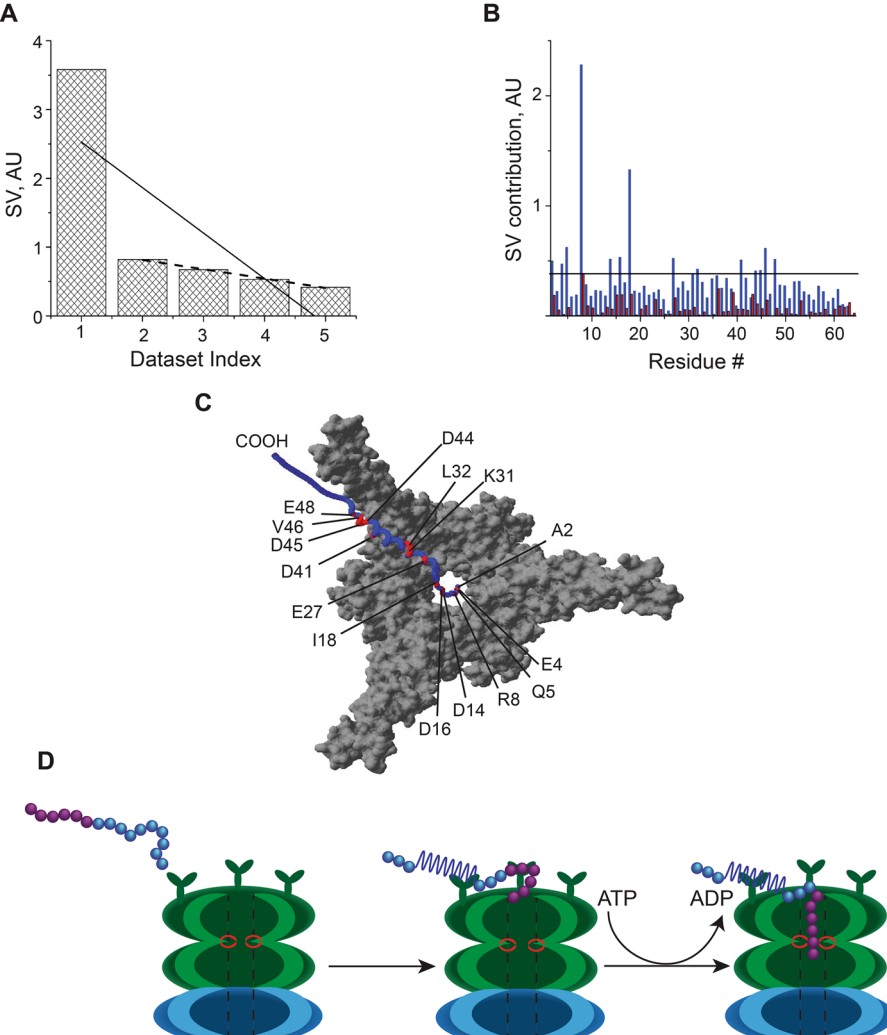

**Fig. 5 Functional Pup–Mpa interaction in metabolically active cells. a** and **b** Singular values and changes in cross peak intensity for each amino acid upon binding to Mpa for the first (blue) and second (red) binding modes (Supplementary Data 1). The threshold line was set to the maximum contribution of the second binding mode, which represents the average noise of the NMR spectra. Continuous and dashed lines in **a** represent linear fits of SVs with and without first binding mode, respectively. **c** Residues comprising the first binding mode (red) between Pup (blue) and Mpa (gray) are mapped onto a Pup–Mpa complex (PDB 3M9D)[39]. **d** The mechanism of Pup threading into the Mpa cavity can be observed in-cell using the improved bioreactor.

## Discussion

It is generally understood that the metabolome dramatically affects protein–protein interactions[5]. The activities of many metabolic enzymes and their complexes are regulated by allosteric inhibitors and enhancers that are present at defined steady state concentrations[41]. These concentrations are notoriously difficult to mimic in solution since the physiological concentrations of most metabolites are far from chemical equilibrium. This is especially true in bacteria, which do not posses a robust ATP buffering system, and in which ATP cellular concentration is determined solely by a combination of ATP synthesis and degradation[42,43]. Real time in-cell NMR allows us to maintain physiological concentrations of metabolites to study the structure-function relationships that govern protein–protein interactions. The improvements in bioreactor technology described here can maintain high adenylate energy charge in E. coli enabling functional protein–protein interactions to be monitored in real time.

## Methods

**Cell growth**. Fifty milliliters of Luria broth (LB) supplemented with 150 μg/mL of carbenicillin and 50 μg/mL of kanamycin was inoculated with a single colony of E coli strain HI-Control BL21(DE3) transformed with pASK-Pup and pRSF-Mpa[33] and grown overnight at 37 °C. The overnight culture was transferred to 500 mL of LB containing 150 μg/mL of carbenicillin and 50 μg/mL of kanamycin and grown to an $OD_{600}$ of 0.7–1.0. The culture was centrifuged at $200 \times g$ for 20 min and prepared for casting in alginate.

**Cell casting**. Pelleted cells (~500 μL) were mixed 1:1 (v/v) with a 2% alginate solution in hybrid growth medium salts, HGM salts, 50 mM HEPES, pH 7.5, 2 mM $CaCl_2$, 0.8 mM $MgSO_4$, 5.3 mM KCl, 110 mM NaCl, and 3 mM $NaH_2PO_4$. The alginate cell suspension was loaded into a 3 mL syringe fitted with a Luer-Lok tip connected to a 40 mm length of tygon tubing (I.D. 0.79 mm) with a blunt 21 gauge needle fixed on the end and injected into an atomizer by using a syringe pump (New Era Pump Systems NE-300) at a rate of 300 μL/min. The atomizer consisted of a pipette through which compressed air flowed at a rate of 5.5 liters per minute (Fig. 2a). The small orifice created a mist that was centered over 25 mL of 150 mM $CaCl_2$. As the alginate/cell mixture was injected into the atomizer stream, the mixture was uniformly dispersed into the $CaCl_2$ solution allowing the $Ca^{2+}$ to polymerize the alginate and encapsulate the cells within the beads. The $CaCl_2$ solution was decanted off of the beads and replaced with 25 mL of HGM, HGM salts supplemented with 4 g/L glucose, 1 mg/mL $NH_4Cl_2$, 1 mg/mL ISOGRO (Sigma Aldrich) and 1 mg/mL thiamine.

**Determination of average bead size**. Prior to placing cell/alginate beads in bioreactor, 3 sets of four beads were lined up with a razor blade and the total length measured with calipers (Husky). The measurement was divided by four to yield an average bead diameter and uncertainty for each set of beads. Following 24 h of incubation in the bioreactor the beads were removed and re-measured. Before and after magnified images of the beads were captured by using an Evos FL cell imaging system (Thermo Fisher).

Minimal size of the beads to resist the hydrodynamic drag can be calculated by equating the gravity force that acts on a spherical bead suspended in medium to Stoke's drag, $4\pi R_{min}^3 (d_A - d_w)g = 6\pi R_{min} \eta v$[44], where $R_{min}$ is the radius of the bead, $d_A$ and $d_w$ are the densities of alginate beads and water, respectively, $g$ is the gravity constant, $\eta$ is the viscosity, and $v$ is the speed of flow. Since $v = J/A$, where $J$ is the flow rate and $A$ is the total cross-sectional area of the flow, $R_{min} = (3\eta J/(A(d_A - d_w)g))^{1/2}$. Assuming that $\eta = 0.89 \times 10^{-3}$ Pa × s[44], $J = 80$ μL/min, $d_A - d_w = 60$ kg/m³[45], and $A \approx 1 \times 10^{-7}$ m², $R_{min} \approx 0.3$ mm. The average radius of the beads used in this work was ≈ 0.4 mm, which is larger than $R_{min}$.

**Bioreactor setup**. To ensure an uninterrupted flow of medium the bioreactor employed a standard screw-cap NMR tube with a PTFE/silicone septum to create a seal around the inlet and outlet tubing. Starting the flow of medium with a gravity siphon provides the bioreactor with a means to prime the system, as well as to eliminate bubbles from the bioreactor that obstruct the flow. A peristaltic pump was used to maintain a flow rate of 80 μL/min by acting as a brake on the native flow rate resulting from the height difference between the inlet and outlet reservoirs.

A horizontal drip irrigation stem was made from 3 lengths of plastic microhematocrit capillary tubes (Globe Scientific) connected by 3D-printed (Dremel Model 3D45) 10 mm joints and bonded together with plastic bond (Loctite). A 13 mm length of ultrahigh molecular weight, UHMW, polyethylene porous rod (Scientific Commodities) with an O.D. of 1.55 mm and an 8 mm long channel through the center with a diameter of 0.85 mm was fixed to the end of the rod with plastic bond (Loctite). Fabricating the stem out of plastic capillary tubing

provided rigidity and the connecting joints doubled as a dam to contain cells within the sampling volume of the NMR tube. A 50 mm length of Tygon tubing fitted with a Luer-Lok syringe connector was attached to the top of the horizontal drip irrigation stem to connect to the inlet tubing. Waste medium was removed from the bioreactor by using 2 m of PTFE tubing (I.D. 0.5 mm) with a tubing connector (GE Healthcare) that allowed for separation of the irrigation stem and outlet tubing within the bioreactor.

A reservoir containing fresh HGM was placed in a 42 °C water bath (Branson model 2800). A 2 m length of Tygon tubing (I.D. 0.79 mm) weighed down by a Luer-Lok syringe connector was inserted into the reservoir. The tubing was insulated by using a 1.21 m length of 1.25 cm foam pipe insulation (Everbilt) filled with polyester fibers (Poly-fil). E. coli transformed with pASK-Pup or pRSF-Mpa cast into alginate beads were transferred to a 5 mm, 600 MHz screw cap NMR tube with a PTFE/silicone septum (New Era). The irrigation stem and outlet tubing were inserted through the septum. A second Luer-Lok syringe connector was used to connect the inlet tubing to the irrigation stem. A flow rate of 80 μL/min was controlled by a pump (Pharmacia LKB) attached to the waste tubing.

**Expression of [$U$-$^{15}$N] Pup**. Bioreactor overexpression of uniformly labeled [$U$-$^{15}$N] Pup was achieved by flowing [$U$-$^{15}$N] HGM, HGM salts supplemented with 4 g/L glucose, 1 mg/mL [$U$-$^{15}$N] $NH_4Cl_2$, 1 mg/mL [$U$-$^{15}$N] ISOGRO (Sigma Aldrich), 1 mg/mL thiamine, containing 150 μg/mL carbenicillin, and 50 μg/mL kanamycin through the bioreactor while on the bench top with the NMR tube in a 37 °C water bath. Overexpression of Pup was induced with 2 mg/mL of anhydrotetracycline in dimethylformamide to a final concentration of 0.2 μg/mL and allowed to proceed for 18 h.

**Expression of Mpa**. Following overexpression of Pup the bioreactor was transferred to the NMR. [$U$-$^{15}$N] HGM was removed from the fresh medium reservoir and replaced with $^2$D-HGM, HGM prepared in 90% $H_2O$/10% $D_2O$, containing 150 μg/mL carbenicillin and 50 μg/mL kanamycin, in which [$U$-$^{15}$N] $NH_4Cl_2$ was replaced with $NH_4Cl_2$, and [$U$-$^{15}$N] ISOGRO was replaced by 1 g/L of Bacto yeast extract (BD Biosciences) and 1 g/L of Bacto tryptone (BD Biosciences), and allowed to equilibrated for 30 min. Overexpression of Mpa was induced by adding 1 mM isopropyl β-D-1-thiogalactopyranoside.

**Preparation of E. coli lysate containing overexpressed Pup**. Fifty milliliters of LB medium supplemented with 150 μg/mL of carbenicillin was inoculated with a single colony of E.coli strain HI-Control BL21 (DE3) transformed with pASK-Pup and incubated overnight at 37 °C in a rotating shaker. The overnight culture was transferred to 1 L of LB containing 150 μg/mL carbenicillin and grown at 37 °C in a rotating shaker to an $OD_{600}$ of 0.7. Cells were collected by centrifugation at $200 \times g$ for 20 min. The wet cell pellet was transferred to 1 L of minimal (M9) medium containing 1.0 g/L of [$U$-$^{15}$N] ammonium chloride as the sole nitrogen source and 0.2% glucose as the sole carbon source. Overexpression of Pup was induced with 2 mg/mL of anhydrotetracycline in dimethylformamide to a final concentration 0.2 μg/mL and allowed to proceed for 4 h. Cells were collected by centrifugation at $200 \times g$ for 20 min and extensively washed 8 times with 5 mL aliquots of 10 mM phosphate buffer pH 6.8. The washed cells were suspended in 6 mL of 10 mM phosphate buffer and sonicated with a Model 250 Digital Sonifier (Branson). The lysate was centrifuged at $30,000 \times g$ for 45 min at RT and the supernatant was collected and used in subsequent experiments.

**NMR spectroscopy**. All NMR spectra were recorded at 310 K using a 600 MHz Avance III NMR spectrometer equipped with a QCI-P cryoprobe (Bruker). $^{15}$N-edited SOFAST-HMQC spectra were acquired with 256 scans. The spectral widths in the $^1$H and $^{15}$N dimensions were 14 and 31 ppm, respectively and were digitized by 1024 and 64 points in the $^1$H and $^{15}$N dimensions, respectively with a recycling time of 100 ms. The in-cell [$U$-$^{15}$N] Pup spectrum was recorded in the absence of Mpa to establish a reference spectrum. One millimolar isopropyl β-D-1-thiogalactopyranoside was added to the fresh medium reservoir and $^{15}$N-edited SOFAST-HMQC spectra were recorded 5 times in succession over 5 hours.

Proton-decoupled $^{31}$P spectra were recorded for E. coli cells cast in alginate beads with and without the bioreactor. Spectra were recorded at 3.5 h time intervals with a total acquisition time of 25 h for both samples. The $^{31}$P peak intensity at −11.5 ppm that contains contributions from the α-phosphate of both ATP and ADP and diphosphate of NAD⁺ and NAD(H) was integrated. The experiments were performed in duplicate. All spectra were processed with Topspin version 3.2 (Bruker) and analyzed by using CARA software.

To study the effect of AMP-PCP on the Pup–Mpa interaction, 250 μL of E. coli lysate containing overexpressed [$U$-$^{15}$N] Pup was diluted 1:1 with 10 mM phosphate buffer pH 6.8, 90% $H_2O$/10%$D_2O$. Pup–Mpa samples contained 110 μM of purified Mpa, and Pup–Mpa with AMP-PCP samples contained 250 μM AMP-PCP. $^{15}$N-edited heteronuclear single quantum coherence, HSQC, spectra[46] were collected with a spectral width of 14 and 35 ppm in the proton and nitrogen dimensions, respectively. 1024 and 128 points were collected in the proton and nitrogen dimensions, respectively. All spectra were processed using Topspin version 3.2 (Bruker) and analyzed using CARA software.

**Data analysis**. NMR spectral data were analyzed as previously reported[34,47]. Cross peak intensities of the [15]N-edited SOFAST-HMQC spectra of free Pup and Pup–Mpa at different time intervals were determined. Peak intensities were scaled and changes resulting from the expression of Mpa were calculated by using $\Delta I = (I/I_{\text{ref}})_{\text{bound}} - (I/I_{\text{ref}})_{\text{free}}$, where $(I/I_{\text{ref}})_{\text{free}}$ is the scaled intensity of an individual peak in the free Pup spectra, $(I/I_{\text{ref}})_{\text{bound}}$ is the scaled intensity of individual peaks in the Pup–Mpa complex spectra. $I_{\text{ref}}$ is a glutamine peak at 7.52 ppm and 122.2 ppm in the proton and nitrogen dimensions, respectively, that does not shift during expression of Mpa. Data were compiled into matrix M using in Excel (Microsoft) and exported as a ASCII text file that was read by MATLAB (Mathworks). Random matrices of values between −1 and 1 were generated using $A = -1 + 2*\text{rand}(64,5)$ command in MATLAB.

Changes in cross peak intensities of the [15]N-edited HSQC spectra were calculated as a ratio of $(I/I_{\text{ref}})_{\text{bound}}$ over $(I/I_{\text{ref}})_{\text{free}}$, where $(I/I_{\text{ref}})_{\text{bound}}$ represents the peak intensity of Pup bound to Mpa in the presence and absence of AMP-PCP and $(I/I_{\text{ref}})_{\text{free}}$ represents the peak intensity of Pup in the absence of Mpa. Area plots of $\Delta I$ were constructed using Originpro (Originlab)

Singular value decomposition (SVD) on matrix M (Supplementary Table 1 and 2) was executed using the [U, S, V] = svd[M] command. The analysis generated 3 matrices U, S, and V as an output. U represents the left singular vectors, S representing the singular-value matrix, and V representing the right singular vectors. To visualize the contribution to each binding mode, a scree plot of the singular values was generated. Scree plots were fitted by linear regression (Microsoft Excel) to determine the coefficient of determination, $r^2$[48,49]. Experimental noise is described by the magnitudes of the second-order and higher-order binding modes. The threshold for determining the amino acids involved in the changes in quinary interactions was set to the maximum contribution of the second binding mode.

**Mpa calibration curve**. Fifty milliliters of Luria broth (LB) supplemented with 50 µg/mL of kanamycin was inoculated with a single colony of pRSF-Mpa and grown overnight at 37 °C. The overnight culture was transferred to 500 mL of LB containing 50 µg/mL of kanamycin and grown to an $OD_{600}$ of 0.7–1.0. Overexpression of Mpa was induced for 12 h by adding 1 mM isopropyl β-D-1-thiogalactopyranoside. The culture was centrifuged at $200 \times g$ for 20 min at room temperature, resuspended in 20 mL of lysis buffer, 50 mM sodium phosphate, pH 8.0, 300 mM NaCl, 10 mM imidazole, per gram of pelleted cells and sonicated with a Model 250 Digital Sonifier (Branson). The lysate was centrifuged at $30,000 \times g$ for 45 min at RT and the supernatant was collected. Mpa was purified by Ni-NTA affinity chromatography. Stock concentration of purified Mpa was determined by Bradford colorimetric assay. Stock dilutions were electrophoresed on a 10% sodium dodecyl sulfate polyacrylamide gel. Coomassie blue G-250 was used to stain the gel. Precision Plus Protein (BioRad) was used as a molecular size standard. Band intensity was quantified using ImageJ software[50] to generate a calibration curve.

**In-cell Mpa concentrations**. Fifty milliliters of Luria broth (LB) supplemented with 50 µg/mL of kanamycin was inoculated with a single colony of pRSF-Mpa and grown overnight at 37 °C. The overnight culture was transferred to 500 ml of LB containing 50 µg/mL of kanamycin and grown to an $OD_{600}$ of 0.7–1.0. The culture was centrifuged at $200 \times g$ for 20 min at RT and cast in alginate beads. Cells were placed into the bioreactor on the benchtop in a 37 °C water bath to replicate conditions in the NMR spectrometer. Overexpression of Mpa was induced by adding 1 mM isopropyl β-D-1-thiogalactopyranoside. Samples of alginate beads were removed from bioreactor every 2 h for 12 hours and at 24 h and normalized to 33 mg for analysis. Alginate bead samples were suspended in 500 µL of lysis buffer and sonicated with a Model 250 Digital Sonifier (Branson). The lysate was centrifuged at $30,000 \times g$ for 45 min at RT and the supernatant was collected. Mpa was purified from the supernatant by Ni-NTA affinity chromatography spin columns (Qiagen). Purified Mpa was electrophoresed on a 10% sodium dodecyl sulfate polyacrylamide gel. Coomassie blue G-250 was used to stain the gel. Precision Plus Protein (BioRad) was used as a molecular size standard. Band intensity was quantified using ImageJ software[50] and compared to a standard curve generated by using purified Mpa to calculate in-cell Mpa concentrations. Because recovery of Mpa from the affinity column may not be 100%, the concentrations measured represent a lower limit to what is present in-cell.

**Binding curve**. Mpa binding curves were constructed by plotting the estimated intracellular concentration of Mpa against the intensity of I18 located at 7.934 ppm and 122.046 ppm [1]H and [15]N dimensions, respectively. The isotherm was fit to a single site binding equation, $Y = B_{\max} \times X/(K_d + X)$, using Prism 6 (Graphpad) software, where $B_{\max}$ is the maximum value of Y, X is the concentration of Mpa in µM, and $K_d$ is the equilibrium dissociation binding constant, in µM.

**Statistics and reproducibility**. Plotted values are calculated as the mean ± standard deviation with individual data points shown in dot-plot format to present data distribution.

**Bioreactor images**. Images of the diffuser and bioreactor were collected using a Canon Powershot SX50 HS camera.

**Reporting summary**. Further information on research design is available in the Nature Research Reporting Summary linked to this article.

## Data availability
The authors declare that the data supporting the findings of this study are available within the article, Supplementary Information, and Supplementary Data files. All source data underlying the graphs are available as Supplementary Data 1, 2, and 3.

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

## Acknowledgements

The work was supported by NIH grant R01GM085006 and P01HL1463678431 to A.S.

## Author contributions

L.B. performed, L.B., D.S.B., and A.S. analyzed the experiments, L.B, D.S.B., and A.S. wrote the main manuscript text, and A.S. concieved the experiments.

## Competing interests

The authors declare no competing interests.
