## [Peer Review File · Communications Biology]

Reviewers' comments:

Reviewer #1 (Remarks to the Author):

The manuscript by Shekhtman and co-workers reports on an interesting development of their previous bioreactor setup for in-cell NMR. The research reported is interesting and deserves publication. The authors should, however, perform the following modifications to figure 1, before the quality of the presentation matches a publication standard.

Move figure S1 to a panel of figure 1. The sketch of the magnet is redundant, so it could be removed. On the other hand, a measure of the relative heights of the components is necessary, as a gravity siphon effect is used. The photo in figure 1A is not of sufficient quality (illumination should be improved), and a photo of the assembled UHMW polyethylene diffuser should be provided. Finally, still regarding the scheme: in our hands, a peristaltic pump is not sufficient as overfull protection, because it needs the tubes to be constantly full. How is this coped with?

Reviewer #2 (Remarks to the Author):

In this manuscript the authors describe an improved design of NMR bioreactor and its application to monitor a protein-protein interaction, that the same group had previously characterized under static conditions, by time-resolved STINT NMR in bacteria. The improved design allows a substantial increase in cell metabolic activity, compared to the previous iteration. Active *E. coli* is shown to maintain higher intracellular ATP levels and this in turn affects the outcome of the interaction experiment. Additional residues of the investigated protein appeared to interact with the proteasome, compared to previous experiments, and that is attributed to an ATP-dependent process previously described.

Overall, this work is an incremental but interesting improvement of the authors' bioreactor setup, and offers a further example of the potential of in-cell NMR approaches to monitor biological event as they occur in real time.

Some minor issues must be addressed before publication:

1) In the introduction, one has the impression that the ~40% loss in energy production is a general limitation of bioreactors: "Current bioreactor technology can sustain metabolic energy levels for up to 24 h with ~40% loss of...". It should be made clear that this only refers to the authors' previous design.

2) The introduction diffusely uses the general term 'cells', and this is correct when describing the general viability requirements of in-cell NMR. However, in the "Current bioreactor technology..." paragraph it should be made clear that this bioreactor was specifically applied to *E. coli*, as other works are cited where other types of cells were used. The ~40% loss in ATP levels also refers to *E. coli*?

3) At page 4, "re-examined using an improved bioreactor that simplifies the delivery of labeled proteins into cells", how can a bioreactor simplify the delivery of labeled proteins into cells? Also, protein delivery was not performed in this work.

4) The above sentence should be rephrased to make clear that the improvement refers to the previous authors' design.

5) in the first paragraph of the Results, "Under traditional in-cell NMR conditions, high numbers of cells are suspended...", what is 'traditional' in in-cell NMR? Other cell types are not kept in suspension. This should be rephrased as something like: "Under static in-cell NMR conditions, high numbers of bacterial cells are suspended..."

6) At page 5: "Unlike agarose threads, alginate expands as cells grow", if this was observed in this setup it would be useful to show the data. Previous work reporting this could also be cited.

7) At page 4: "A schematic of the improved bioreactor is shown in Figure 1A ", but only the cell casting setup is shown. It would also be useful to include an actual photo of the empty bioreactor, where the horizontal drip irrigation stem (is this actually present?) and the UHMW porous rod can be seen.

8) In Figure 1B, the ^{31}P signal at -6 ppm likely arises from overlapped γ -ATP and β -ADP, as found in the literature. Are the two spectra processed with the same parameters?

9) In figure 1C, are error bars standard deviation ($n=2$), or S.E.M. obtained from signal integration? They are clearly too small compared to the oscillations between points. Maybe actual data from the duplicate experiments could be shown in the graph, instead of the mean values?

10) In the discussion (or elsewhere), it could be useful to explain how the UHMW porous rod increases the efficiency of the bioreactor versus a simple inlet tube opening at the bottom of the tube.

11) In the methods, in the description of the cell casting device, what kind of atomizer was used?

Reviewer #3 (Remarks to the Author):

The authors describe the improvements in bioreactor technology for in-cell NMR spectroscopy in order to sustain high levels of metabolites such as ATP (>95% for up to 24 hour) sufficient for protein expression and vigorous cell growth. By using this improved system, the interactions between prokaryotic ubiquitin-like protein, Pup, and mycobacterial proteasomal ATPase, Mpa was identified. The identified interactions included a set of residues encompassing the N-terminus of Pup, which was previously unobserved both in vitro and in metabolically inactive cells. It is well written, has important clinical message, and should be of great interest to the readers. Therefore, I strongly recommend publication.

However, before the publication, the following points need to be addressed.

1. On page 12, lines 5-6 up, the authors discussed that "The lack of visibility of the N-terminal interaction in the previous studies^{34, 35} was due to insufficiently high levels of ATP within the cells during traditional in-cell NMR." If so, *in vitro* NMR interaction analysis in the presence of high concentration of ATP or its slowly hydrolyzed analog ATP γ S could detect the interaction. The author needs to discuss and address more about the possible reason of this difference such as other metabolite, molecular crowding or its combination.
2. On page 17, the authors described the method for quantifying in-cell Mpa concentration in *E. coli* cells with overexpressed Pup. It basically quantified Ni-NTA affinity-purified Mpa from *E. coli* cells rather than directly quantified intracellular Mpa. I suspect that the recovery of Ni-NTA affinity spin column is not perfect, thus the in-cell Mpa concentration is supposed to be underestimated.

3. Figure 3. Was the concentration of overexpressed Pup kept constant during overexpressing Mpa? Was it confirmed for example by SDS-PAGE of whole *E. coli* cells or their extracts?
4. More detailed explanation with figures of experimental set-up of the improved bioreactor system especially in terms of the improvement of an inlet tubing would be of great help to the readers for their reproduction of the bioreactor system.

We thank our editor and reviewers for the comments.

Editor:

While all reviewers find this study interesting, they made suggestions to improve the data presentation and to clarify some issues. To address the reviewers' concerns, we ask you to please show that alginate expands as cells grow and to validate that the concentration of overexpressed Pup was kept constant during overexpressing Mpa.

The average diameter of the alginate beads was measured before being placed in the bioreactor and after 24 h incubation in the bioreactor. The data is presented in Supplementary Figure 2. We also referenced previous work showing that Pup concentration remains constant during Mpa overexpression. (Results: Functional Protein-protein interactions, Paragraph 2)

Reviewer 1:

1. *The authors should, however, perform the following modifications to figure 1, before the quality of the presentation matches a publication standard.*

a. *Move figure S1 to a panel of figure 1. The sketch of the magnet is redundant, so it could be removed. On the other hand, a measure of the relative heights of the components is necessary, as a gravity siphon effect is used.*

We moved Figure panel S1A to Figure 1. The peristaltic pump eliminates the need for a height difference to drive a gravity siphon, so no component heights are given.

b. *The photo in figure 1A is not of sufficient quality (illumination should be improved).*

The photo of the thin slice of the UHMW polyethylene porous polymer was magnified and moved to Supplementary Figure 1.

c. *A photo of the assembled UHMW polyethylene diffuser should be provided.*

We modified Figure 1 to include an enlarged image of the diffuser tip, and the assembled and disassembled bioreactor.

2. *Finally, still regarding the scheme: in our hands, a peristaltic pump is not sufficient as overfull protection, because it needs the tubes to be constantly full. How is this coped with?*

With inlet and outlet reservoirs on the floor a break in the inlet or outlet lines will result in solution flowing back into the inlet reservoir with no flooding of the probe.

Reviewer 2:

1. *In the introduction, one has the impression that the ~40% loss in energy production is a general limitation of bioreactors: “Current bioreactor technology can sustain metabolic energy levels for up to 24 h with ~40% loss of...”. It should be made clear that this only refers to the authors’ previous design.*

It was clarified that this statement referred to our previous design. (Intro Paragraph 3).

2. *The introduction diffusely uses the general term ‘cells’, and this is correct when describing the general viability requirements of in-cell NMR. However, in the “Current bioreactor technology...” paragraph it should be made clear that this bioreactor was specifically applied to E. coli., as other works are cited where other types of cells were used. The ~40% loss in ATP levels also refers to E. coli?*

It was stated that the viability requirements specified refer to E. coli cells. (Intro Paragraph 3)

3. *At page 4, “re-examined using an improved bioreactor that simplifies the delivery of labeled proteins into cells”, how can a bioreactor simplify the delivery of labeled proteins into cells? Also, protein delivery was not performed in this work.*

This statement was deleted. (Intro Paragraph 5)

4. *The above sentence should be rephrased to make clear that the improvement refers to the previous authors’ design.*

It was clarified that this statement referred to our previous design. (Intro Paragraph 5)

5. *In the first paragraph of the Results, “Under traditional in-cell NMR conditions, high numbers of cells are suspended...”, what is ‘traditional’ in in-cell NMR? Other cell types are not kept in suspension. This should be rephrased as something like: “Under static in-cell NMR conditions, high numbers of bacterial cells are suspended...”*

The statement was changed to “During in-cell NMR experiments in the absence of a bioreactor”. (Results paragraph 1)

6. *At page 5: “Unlike agarose threads, alginate expands as cells grow”, if this was observed in this setup it would be useful to show the data. Previous work reporting this could also be cited.*

The average diameter of the alginate beads was measured before being placed in the bioreactor and after a 24 h incubation in the bioreactor. The data is presented in Supplementary Figure 2.

7. *At page 4: “A schematic of the improved bioreactor is shown in Figure 1A “, but only the cell casting setup is shown. It would also be useful to include an actual photo of the empty bioreactor, where the horizontal drip irrigation stem (is this actually present?) and the UHMW porous rod can be seen.*

The image of the cell casting setup was moved to Figure 2. We prepared a new Figure 1 showing the schematic of the improved bioreactor, an enlarged image of the diffuser tip, and the assembled and disassembled bioreactor.

8. *In Figure 1B, the 31P signal at -6 ppm likely arises from overlapped γ -ATP and β -ADP, as found in the literature. Are the two spectra processed with the same parameters?*

Yes, we indicated that the same processing parameters were used and indicated that there is the possibility of overlap.

Figure 1 was moved to Figure 2 and modified to include β -ADP.

9. *In figure 1C, are error bars standard deviation (n=2), or S.E.M. obtained from signal integration? They are clearly too small compared to the oscillations between points. Maybe actual data from the duplicate experiments could be shown in the graph, instead of the mean values?*

Figure 2C was modified to indicate that the experiment was performed in duplicate and that the error bars are SEM.

10. *In the discussion (or elsewhere), it could be useful to explain how the UHMW porous rod increases the efficiency of the bioreactor versus a simple inlet tube opening at the bottom of the tube.*

The UHMW porous rod allows us to increase the effective surface of the orifice as compared to a simple inlet tube opening at the bottom of the NMR tube. This in turn decreases hydrodynamic drag, which perturbs the beads. We expanded on the effectiveness of the diffuser in the Improved Bioreactor section of Results and in Paragraph 2 of the Results. We also included calculation of optimal bead size to minimize hydrodynamic drag. (Methods: Determination of average beads size)

11. *In the methods, in the description of the cell casting device, what kind of atomizer was used?*

We used an atomizer based on compressed air passing through small orifice and creating an atomized “mist”. This process was described more clearly in the Cell Casting section of Methods.

Reviewer 3:

1. *On page 12, lines 5-6 up, the authors discussed that “The lack of visibility of the N-terminal interaction in the previous studies^{34, 35} was due to insufficiently high levels of ATP within the cells during traditional in-cell NMR.” If so, in vitro NMR interaction analysis in the presence of high concentration of ATP or its slowly hydrolyzed analog ATP γ S could detect the interaction. The author needs to discuss and address more about the possible reason of this difference such as other metabolite, molecular crowding or its combination.*

High ATPase activity of Mpa, $k_{cat} \sim 70 \text{ min}^{-1}$ (Delley et al; J Biol Chem. 2012 Mar 9; 287(11): 7907–7914), precludes us from using ATP in this experiment. The HSQC spectra of the Pup-Mpa interactions was collected in the presence of a non-hydrolyzable ATP analog β, γ -methyleneadenosine 5'-triphosphate, AMP-PCP. The results of the experiment are discussed in the last paragraph of the results and the data shown in Supplementary Figure 5.

2. *On page 17, the authors described the method for quantifying in-cell Mpa concentration in E. coli cells with overexpressed Pup. It basically quantified Ni-NTA affinity-purified Mpa from E. coli cells rather than directly quantified intracellular Mpa. I suspect that the recovery of Ni-NTA affinity spin column is not perfect, thus the in-cell Mpa concentration is supposed to be underestimated.*

We indicated that the resolved concentrations represent a lower limit to the in-cell concentration. (Methods: In-cell Mpa concentrations)

3. *Figure 3. Was the concentration of overexpressed Pup kept constant during overexpressing Mpa? Was it confirmed for example by SDS-PAGE of whole E. coli cells or their extracts?*

We referenced previous work showing that Pup concentration remains constant during Mpa overexpression. (Results: Functional Protein-protein interactions Paragraph 2)

4. *More detailed explanation with figures of experimental set-up of the improved bioreactor system especially in terms of the improvement of an inlet tubing would be of great help to the readers for their reproduction of the bioreactor system.*

We prepared new Figure 1 showing the schematic of the improved bioreactor, an enlarged image of the diffuser tip, and the assembled and disassembled bioreactor. The image of the cell casting setup was moved to Figure 2. We clarified that the atomizer was compressed air passing through a small orifice, creating an atomized “mist” in the Cell Casting section of Methods.

REVIEWERS' COMMENTS:

Reviewer #1 (Remarks to the Author):

The authors have satisfactorily addressed my questions.

Reviewer #2 (Remarks to the Author):

In the revised version of the manuscript, the authors have adequately addressed all my comments, improving the clarity and overall quality of the work. Therefore, I recommend publication in Communications Biology.

Reviewer #3 (Remarks to the Author):

Corrections suggested by editor and reviewers has been completed by the authors and found satisfactory.